# Factors Affecting Adverse Health Effects of Gasoline Station Workers

**DOI:** 10.3390/ijerph181910014

**Published:** 2021-09-23

**Authors:** Umakorn Tongsantia, Sunisa Chaiklieng, Pornnapa Suggaravetsiri, Sari Andajani, Herman Autrup

**Affiliations:** 1Dr. PH Program in Public Health, Faculty of Public Health, Khon Kaen University, Khon Kaen 40002, Thailand; umato@gmail.com; 2Department of Environmental Health and Occupational Health and Safety, Faculty of Public Health, Khon Kaen University, Khon Kaen 40002, Thailand; 3Department of Epidemiology and Biostatistics, Faculty of Public Health, Khon Kaen University, Khon Kaen 40002, Thailand; porsug@kku.ac.th; 4School of Public Health and Interdisciplinary Studies, Auckland University of Technology, Auckland 1142, New Zealand; sari.andajani@aut.ac.nz; 5Institute of Public Health, Aarhus University, 8000 Aarhus, Denmark; ha@ph.au.dk

**Keywords:** gasoline station, tt-muconic acid, benzene, risk factor, adverse symptom

## Abstract

This cross-sectional study examined the risk factors affecting adverse health effects from benzene exposure among gasoline station workers in Khon Kean province, Thailand. An interview questionnaire of adverse symptoms relating to benzene toxicity was administered to 151 workers. Area samplings for benzene concentration and spot urine for tt-muconic acid (tt-MA), a biomarker of benzene exposure, were collected. The factors associated with adverse symptoms were analysed by using multiple logistic regression. It was found that these symptoms mostly affected fuelling workers (77.5%), and the detected air benzene reached an action level or higher than 50% of NIOSH REL (>50 ppb). The top five adverse symptoms, i.e., fatigue, headache, dizziness, nasal congestion, and runny nose, were reported among workers exposed to benzene. More specific symptoms of benzene toxicity were chest pain, bleeding/epistaxis, and anaemia. The detected tt-MA of workers was 506.7 ug/g Cr (IQR), which was a value above the BEI and higher than that of asymptomatic workers. Risk factors significantly associated with adverse symptoms, included having no safety training experience (OR_adj_ = 5.22; 95% CI: 2.16–12.58) and eating during work hours (OR_adj_ = 16.08; 95% CI: 1.96–131.74). This study urges the tightening of health and safety standards at gasoline stations to include training and eating restrictions while working in hazardous areas.

## 1. Introduction

Recently, there has been an increase in the ownership of registered cars in Thailand. Between 2015 and 2018, car ownership increased by an average of 3.89% annually [1], and the sales figures of gasoline stations from 2012–2015 showed that gasoline sales had increased from previous years [2]. Although, benzene found in the gasoline mix has been set at no more than 1.1% by volume since 2012 [3], which was a significant decrease from the previous figure of 3.8% by volume, it continues to cause health concerns. Studies show that the levels of benzene released from vehicles are still higher compared to those found in ambient air and indoors [4]. Benzene released from fuel vapours can be harmful when inhaled, especially among people working in gasoline stations [5,6,7].

The primary substances found in gasoline are organic compounds with low boiling points and high vapor pressure, such as volatile organic compounds or VOCs, benzene toluene ethylbenzene xylene, or BTEX—[8]. The most common symptoms reported in gasoline station workers were headache, fatigue, throat irritation, nose irritation, nausea, dizziness, and depression [9]. Most occupational and harmful exposure to benzene can be identified in various ways, for example, the analysis of benzene concentration in the working ambient air compared with the occupational exposure limit (OEL) 0.1 ppm [10].

In Thailand, previous findings of workers at gasoline stations showed that exposure to benzene during the participants’ working period was lower than the OEL [11]. Another analysis involved the trans-, trans-muconic acid (tt-MA), a biomarker of exposure, which is detected by collecting workers’ urine at the end of their shift [12]. One recent study found abnormalities in the body’s organs relating to various types of symptoms of benzene toxicity among gasoline station workers in Thailand [13]. Mild symptoms of benzene exposure, include irritation of the respiratory system, skin, and eyes. Prolonged exposure affects the nervous system, causing headaches, dizziness, fatigue, etc., and also affects blood circulation, causing symptoms, such as bleeding spots, epistaxis, and leukaemia [14,15]. In another case study on symptoms reported in the previous three months, workers’ exposure to benzene were reviewed and their symptoms were found to vary from those at the mild level, such as drowsiness, dizziness, headaches, and tremors, compared to those at the high level, such as unconsciousness and cancer. Developing cancer from benzene exposure has also been reported by International Agency for Research on Cancer (IARC) (acute myeloid leukemia, acute lymphocytic leukemia, multiple myeloma, and non-Hodgkin lymphoma) [16].

The severity levels of symptoms relating to benzene toxicity that are experienced by gasoline station workers depend on various factors. A previous study found that the factors affecting benzene exposure, identified by tt-MA detection, include gasoline station workers standing close to the fuel dispenser during their service period, receiving no relevant training, and working in a location where the benzene concentration is greater than 50 ppb [17]. The health effects of benzene exposure and their associated factors are still unclear. Therefore, this study aimed to investigate the factors associated with adverse health effects experienced by gasoline station workers in Thailand.

## 2. Materials and Methods

### 2.1. Sample Size

The study participants were 151 workers from 41 of the gasoline stations in Khon Kaen province, who were chosen by following the logistic regression sample size calculation [18]. The reference rate from the study of factors affecting urinary tt-MA detection among benzene exposed workers at gasoline stations was a tt-MA level of more than 10% biological exposure index (BEI) in persons who had no job training (Odd ratio = 2.74) [17]. The desired level of confidence was 95% (α = 0.05), at a power of 95% (ß = 0.05). The minimum required sample size was 151 workers. Three specified areas of gasoline stations for data collection were determined. The urban area, where the gasoline stations were between 2 and 5 kilometres away from Mittraphap Road of the Nai mueang sub-district, Khon Kaen Province [15]. These gasoline stations operated from 5 am to 10 pm, and 7 days a week (2 shift working for 8–10 hrs per shift). The Mittraphap road is one of the busy main roads in the area for suburban, the gasoline stations, included in this study, were less than 2 kilometres from the main Mittraphap road which were operated 24 h (3 shifts working for 7–8 hrs per shift). Unlike those gasoline stations in the urban and suburban area, the ones in the rural areas would close around 8 pm (2 shift working for 7–8 hrs per shift), at the owner’s discretion as the rural farmers were likely to end their activities early in the day.

Purposive sampling included 3–4 workers from each determined station. This study was approved by the human ethics committee of Khon Kaen University (No. HE612030). Before entering the study, participants were given a consent form to sign. They had to volunteer to participate, have experience working at a gasoline station for longer than three months, not pregnant or menstruating. Smoking was excluded and drinking alcohol and consumption of food containing sorbic acid were also not allowed 24 h before the day of urine collection.

### 2.2. Questionnaire

The interview questionnaire from the previous study [15] was validated and used in this study. Participants were interviewed by the researcher about the symptoms they had experienced in the past three months. The questionnaire also asked for participants’ demographic characteristics and work-related information, such as length of employment, job function, and working hours.

### 2.3. Adverse Health Symptoms Related to Benzene Toxicity

Adverse health symptoms experienced by workers were classified into five levels [14] as follows; (1) non-symptomatic with smell recognition; (2) mild or low level of symptoms, which were exhaustion, headache, dizziness/fatigue, red eyes/burning eyes/itchy eyes, nasal congestion, sore throat/dry throat, suffocation, cough/hoarseness, runny nose, skin itching/dry skin, cracked skin, itchy skin/red rash/blisters, anorexia, and palpitations; (3) moderate level of symptoms, which were numbness, depression, confusion, blurred vision, insufficient/abnormal breathing, chest pain, nausea and vomiting, burning pain/swelling/muscle weakness, tremor, cramp, scurvy/bleeding, epistaxis, and petechia; (4) high level of symptoms, which were anaemia/epilepsy, convulsions, and unconsciousness; and (5) very high-level symptom, which was leukaemia. High and very high levels of symptoms must be diagnosed by a physician.

### 2.4. Air Sampling

The benzene concentration in the working ambient air was based on active area monitoring. Samples were collected by active area sampling over four hours with coconut charcoal sampling tubes installed at the breathing zone level height of 1.5 metres and kept in containers at a temperature of around 4 °C before they were sent to the laboratory for analysis. The samples were analysed by Gas Chromatography—Flame Ionization Detector (GC-FID) (Limit of detection (LOD) < 0.34 ppb). The analysis lab we use the standard certified lab (Ref No. 0303/17133) [19]. The sampling point for 2–3 points per station as follows: (1) The point near gasoline dispenser (service point); (2) waiting service area (including near dispensers); and (3) cashier booth or cashier working area (Figure 1). The benzene concentration was calculated as an eight-hour time-weighted average of the working period to compare it to the standard regulation of occupational exposure limit (OEL) regulated by National Institute for Occupational Safety and Health (NIOSH-REL) (100 ppb) [10].

### 2.5. Detection of tt-MA in Urine after Work

The day before urine collection, the correct method of collection was explained to workers, i.e., washing hands before collecting the urine and using only urine from the midstream. A 30 mL amount of spot urine was collected at the end of shift work in a jar with a closed lid, stored at a temperature of about 4 °C, and sent to the laboratory for analysis. The concentration of tt-MA was analysed by high-performance liquid chromatography (HPLC) with a ultraviolet (UV) detector with a limit of detection (LOD) of <0.001 mg/g creatinine (CRN) and limit of quantitation (LOQ) of <0.001 mg/g CRN). The tt-MA 97.0% and internal standard vanillic acid 97.0% were used for tt-MA analysis.

### 2.6. Statistical Analyses

STATA version 10.0 software (StataCorp LLC: College Station, TX, USA) was used to analyse the data. The Chi-square and Fisher’s exact tests were used to examine symptoms experienced by workers who had been exposed to benzene >50% OEL (>50 ppb) and those who had a detected tt-MA > BEI (>500 µg/g CRN), at *p*-value ≤ 0.05 of significance. The Spearman Rank test examined the correlations between the biological and environment monitoring levels at *p*-value ≤ 0.05 of significance. The multiple logistic regression was used to analyse factors associated with various levels of adverse health symptoms (moderate to severe/high).

The continuous variable was divided into a dichotomous variable as suggested in the previous study [20]. These continuous variables included age, body mass index (BMI), working experience, previous occupation exposure to benzene, and income before being included in the bivariate analysis. The Shapiro-Wilk test was then used to examine the model of distribution (*p*-value ≤ 0.05). The Fisher’s exact test analysed the correlation between tt-MA and the adverse effect level of benzene concentrations.

The variables measuring working hours were classified into two groups: working hours per day and working days per week. The BMI was classified into four groups: normal, underweight, overweight, and obese. The association between the independent variables and symptom level of benzene exposure was analysed by univariate analysis. Factors with a *p*-value of less than 0.25 [20] from univariate analysis were selected to be candidate variables in the multiple logistic regression analysis. The confounding factors of gender, age, and education were included in the analysis. The odds ratio (OR), adjusted odds ratio (OR_adj_), and 95% CI of OR or OR_adj_ were presented at a *p*-value ≤ 0.05 to indicate the statistical significance of the associated risk factors affecting workers at the gasoline stations.

## 3. Results

### 3.1. Characteristics of Gasoline Station Workers

The participants included 56 male workers (37.1%) and 95 female workers (62.9%). Of these workers, 117 were fuelling workers and 34 cashiers.

Their average age was 34 ± 9.9 years (min–max: 19–67). Most of them were older than 30 years (*n* = 90; 59.6%). Many of the workers had completed secondary school (*n* = 60; 39.7%), followed by those who had completed high school and primary school (*n* = 35, 23.2% and *n* = 27, 17.9%, respectively). More than half of participants were single (50.3%) and 42.4% (*n* = 64) were married. The remaining 7.3% were either divorced or separated. The average income per month was 10,634.5 ± 3407.0 Thai Baht (ranging between 9000 and 10,000 Thai Baht/300 US dollar). Nearly 40% (*n* = 60) of them had been employed for more than two years, with an average working experience of 3.40 ± 4.80 years (ranging from three months to 30 years).

More than half of participants worked in the suburban area (54.30%), followed by those working in the rural or urban areas. Most participants (*n* = 117, 77.5%) worked on a day shift, and 64.9% worked six days per week. Some workers worked seven days per week with no day off. The average working time was 8.95 ± 1.12 h per day. Many workers had working hours of more than eight hours per day.

### 3.2. Benzene Exposure According to Air Benzene and tt-MA Detection

Based on the air benzene concentration ranged from non-detectable to 136.0 ppb (min–max); comparing it with the standard NIOSH-REL (100 ppb) or OEL [10], two workers had been being exposed to benzene in ambient air that exceeded the standard OEL (100 ppb). Three workers were exposed to air benzene above 50 ppb (50% OEL).

The tt-MA detected in the participants was very high, with a maximum of 5986.44 µg/g CRN, an IQR (25th–75th percentile) of 521.19 (86.13–607.32 µg/g CRN) and the 95th percentile of 1384.31 µg/g CRN. A comparison of the tt-MA level with the biological exposure index (BEI) of 500 µg/g CRN, recommended by American conference of governmental industrial hygienist (ACGIH) [12], found that nearly one-third of workers (*n* = 41, 27.2%) showed levels exceeding the BEI. Comparing the fuelling versus the cashier workers, 35 fuelling workers (29.9%) had a tt-MA level exceeding the BEI compared to 6 (17.7%) cashier workers. The highest tt-MA detection value of 5986.44 µg/g CRN was found in fuelling workers, which was fourfold of those found among the casher (maximum detected tt-MA of 1444.74 µg/g CRN).

When comparing to the 50% BEI, it was found that the detected tt-MA level in those with low symptoms (median: 751.82 ug/g CRN) was higher than those who were non-symptomatic. The IQR (25th–75th percentile) of the participants with low symptoms was 826.7 (385.48–1212.18 ug/g CRN) or twice higher than those who were non-symptomatic of 492.89 (372.56–865.46) ug/g CRN.

The participants who reported moderate symptoms, had a detected tt-MA concentration (IQR = 625.65 ug/g CRN (25th–75th: 258.06–1243.84 ug/g CRN), which was lower than those reported high symptoms (IQR = 935.30 ug/g CRN; 25th–75th: 425.81–1361.11 ug/g CRN).

There was a positive correlation between the level of the environmental exposure to benzene and the tt-MA level found in the urine. The participants who had been exposed to an ambient air environment of more than 50% OEL (50 ppb) had a higher tt-MA level than those who worked in the ambient air environment of <50 ppb (median different tt-MA was 432.80 µg/g CRN, *p*-value = 0.015).

### 3.3. Adverse Health Effects of Gasoline Station Workers

Of the 151 participants, there was 90 (59.6%) reported having symptoms associated with benzene exposure. Sixty-one participants experienced no adverse symptoms from benzene toxicity. The top five reported symptoms were headache, dizziness, exhaustion/fatigue, itchy skin/red rash/blisters, nasal congestion, and sore throat/dry throat. Having runny or feeling suffocated significantly correlated with to benzene > 50% OEL and tt-MA > BEI. Mostly experienced symptoms reported the moderate level were chest pain, insufficient/irregular breathing, and anaemia. A symptom-specific to high benzene exposure was anaemia, which was found in fuelling workers (Table 1).

The exposure of benzene concentrations of 50 ppb (50%OEL-NIOSH) or above correlated with moderate to high levels of adverse symptoms. The symptoms significantly associated with benzene exposure were runny nose (*p* = 0.046) and suffocation (*p* = 0.015). The most frequently reported symptoms reported by nearly half of the workers, with a detected tt-MA level above the BEI in their urine, included headaches, itchy skin/red rash/blisters, and dizziness.

### 3.4. Factors Correlated to Adverse Health Effects of Gasoline Station Workers

From univariate analysis, the significant risk factors related to adverse symptoms, included age ≤ 30 years, no training, eating food by the fuelling pumps, and during their working hours (Table 2). The factors with a *p*-value less than 0.25, including being under 30 years, working in urban areas, shift work, using personal protective equipment (PPE) (fabric/cotton mask), no hand washing, no prior training, and eating food by the fuelling pumps, were included into the multiple logistic regression analysis. The confounding variables included in the analysis were gender, age, and education. The analysis found that workers who did not have any prior training were four times more likely to report high symptoms (ORadj = 5.22; 95% CI; 2.16–12.58, *p* < 0.001) than those with prior training. Workers who ate during working hours were 16 times more likely to report the adverse effects than those who refrained from eating while working as a cashier or refuelling (ORadj = 16.08; 95% CI: 1.96–131.74, *p* = 0.010) (Table 3).

## 4. Discussion

Most of the participants were exposed to benzene concentrations in the working ambient air below the standard of OEL-NIOSH [10]. Some workers continued to be exposed to benzene concentrations higher than 50 ppb or the safety level (50% OEL), according to another study conducted in the same research area [20]. One of the gasoline stations in urban zone was observed to have a concentration greater than the OEL (>100 ppb) allowed by the NIOSH [10]. This confirmed a previous study, which found that gasoline workers are exposed to high benzene concentrations in the ambient air, beyond safety standards [21]. The results of this study show that more experienced workers had accumulative benzene concentrations higher than some workers who had been working less than three years (the data were not on a normality distribution, so we need to consider IQR and 95th percentile). The accumulated concentration of workers who worked in a gasoline station for five year or more is 67.6 ppm (95th percentile). The estimate of benzene concentration exposure accumulation per year had an IQR (25th Percentile–75th Percentile) of 2.89 (0.57–3.47) ppm, while the 95th Percentile is 9.27 ppm. Based on the estimated calculation of benzene exposure per year, this study showed that gasoline station workers had higher exposure to benzene when compare to the previous report [22]. This accumulated exposure could support the relationship of exposure to benzene concentration of 0.5–1 ppm per year and the occurrence of hematopoietic cancer or leukaemia [16].

The biomarker of benzene exposure, tt-MA, detected in the urine, was higher among the fuelling workers than the cashiers. Nearly one-third of study participants had a detected tt-MA level exceeding the BEI recommended by ACGIH (2019) [12], which was slightly higher than those reported previously [14]. Symptoms like runny nose, dizziness, headache, and fatigue, confirmed the recent study on the adverse symptoms related to BTEX exposure among gasoline station workers [23]. Most of worker had the symptoms in mild levels (such as headache, itchy skin, fatigue, dizziness, etc.), which is consistent with a previous study that found workers at gasoline stations can be exposed to the VOCs and BTEX substances. Most acute symptoms are associated with irritation of the respiratory tract and skin [24,25]. However, this study found that some workers with detected tt-MA levels in urine, when tested after work, exceeded the BEI standard, and also showed symptoms of hematopoietic effect (anaemia history). Therefore, we still found adverse health effects from benzene exposure in gasoline station workers.

In addition, sore throat was also a significant symptom of toluene exposure found in an earlier study of the same site [26]. Participants who experienced moderate-to-high levels of symptoms, such as chest pain, irregular breathing, bleeding or epistaxis, and haemolytic anaemia, confirmed the findings in other studies [9,17,27]. Our findings confirm an association exists between the detection of high tt-MA levels in the urine with the severity of the symptoms experienced by the participants. Additionally, the significant symptom of epistaxis, found at the moderate level, had been previously recorded by Geraldino et al. [28]. Although this study predominantly found lower air benzene concentrations than the OEL, there were workers who had detected tt-MA levels higher than the BEI. Those participants with a detected high tt-MA level in urine had been working in the high air ambient benzene concentration areas. The findings of this study suggest that workers with moderate-to-high symptoms were more likely to have higher tt-MA concentrations in their urine than those who reported mild or no symptoms. This finding suggests the possibility of workers experiencing chronic or long-term exposure to benzene.

The participants’ gender, marital status, BMI, comorbidities, and family history of the disease were not significantly associated with symptoms reported. However, the age of participants was associated with the risk of experiencing adverse effects. Participants who were younger than 30 years of age were 2.63 times more likely to report symptoms related to benzene exposure than their older peers. This may be due new and young workers (aged 18–30) receiving no health and safety training before they began work at the gasoline station. These workers, therefore, did not know how to protect themselves from the hazards of benzene exposure.

The working characteristic factors, i.e., zone location, job function, number of workdays, working hours, and years of work experience, were not associated with experiencing adverse symptoms. In contrast to previous studies, this study found that job position was associated with benzene exposure detected tt-MA found in the urine. Those working in the fuelling pumps had high tt-MA than the cashiers. The fuelling workers were more likely to work closer to the fuel dispensers for longer hours than the cashiers [16,18]. Workers in urban areas or the inner-city were at a higher risk of benzene exposure than those working in the suburban area or rural area. The urban areas have also shown higher air ambient benzene concentrations than the suburban or rural areas, as reported by Ciarracca et al. [29]. We previously found that the suburban area close to the highway had higher numbers of fuel service stations with 24 h service, which caused a higher concentration of benzene in the ambient air [11]. However, workers in the suburbs split up to two-to-three shifts per day, resulting in less benzene exposure than other areas, and these areas show higher risk prevention when compared to other areas.

For urban areas, due to poor airflow, building density and subsequent heavy air pollution, the numbers of vehicles in urban areas and surrounding areas increase benzene concentration in the ambient air [30]. The rural workers have limited working hours (8–12 h/day), they often do not work in shifts, resulting in longer exposure to benzene, and increasing the risk of benzene exposure. Other concerns of the health risks of gasoline workers and fire risk suggest serious control by using preventive and safety actions to eliminate sources of BTEX release in the hazardous area around the fuel dispenser, such as VRS (vapor recovery system) installation on nozzles [31].

Having no prior training increased the risk of adverse effects from benzene exposure. Workers who had not received any health and safety training were at a higher risk of developing adverse effects than those who had been trained. This is consistent with a previous study [17], which found that new and young workers (aged 18–30) did not receive any health and safety training. There was a significant correlation between having prior health and safety training and higher exposure to benzene detected tt-MA biomarker in the urine [28]. Employers should provide health and safety training for gasoline station workers. In addition, the enforcement and compliance of health and safety policies, as well as the health protection of workers must be made mandatory for gasoline station owners [32]. Findings from this study suggest that health and safety practices for workers, and a routine health surveillance system are highly needed in this enterprise, as also noted by Rocha et al. [33].

The working behaviours, like regular hand washing, showering after work, or PPE use were not significantly associated with the adverse health effects of benzene exposure. The study participants reported using PPE, i.e., wearing cotton hand coverings and masks to protect against sunlight, which unfortunately, increased the accumulation of benzene in the body. For prevention against exposure to chemical mixtures in petroleum products, suitable PPE should include shoes, gaiters to put over pants, and respirators as cartridges with P100 particulate filters (oil resistant) to purify the inhaled air. Neoprene gloves and nitrile, vinyl, and rubber are recommended for effective protection against skin adsorption of workers exposed to benzene at the fuel storage tank of a gasoline station [32,34].

However, other behaviours, like eating food by the fuelling pumps and eating during working hours, were significantly correlated with experiencing adverse symptoms. The benzene vapours can also contaminate food. We observed that participants also stored or placed their food by their operation site or the refuelling area. In some stations, a water fountain for workers was placed in the middle of the gasoline station. Benzene contamination of food and drinking water, placed close to the fuel dispenser’s area, was reported to be very high [35]. It was common for the participants to take a short mealtime break by the fuelling station, especially during the busy hours. Therefore, employers should control the potentially hazardous area, including prohibiting workers from eating within 8 metres of the fuel dispenser zone [31], following the current labour regulation for a hygienic dining area. This study was limited by the lack of confirmation of health adverse effects for high- and a very-high level (level 4–5) of symptoms, which should be diagnosed by the physician or confirmed laboratory testing.

## 5. Conclusions

This study found that some workers (27.2%) in fuelling gasoline stations in Khon Kaen, Thailand, had been exposed to benzene with tt-MA biomarker detection which exceeded the BEI (>500 µg/g CRN). Three of the study participants had benzene concentrations above the action level of the safety limit (>50 ppb). Those levels were associated with significant mild levels of adverse symptoms of benzene exposure, including fatigue, headache, dizziness, nasal congestion, and runny nose, as well as moderate-to-high symptom levels of benzene-specific toxicity, which included chest pain, bleeding or epistaxis, and anaemia. The risk factors associated with adverse health symptoms, included no prior health and safety training and eating food during working hours close to the fuel dispensing areas. These risky behaviours need to be mitigated. Thailand’s Health and Safety Regulations from the Ministry of Labour restrict eating and drinking during working hours, and recommend the provision of a safe eating space away from the hazardous area for gasoline station workers. Regular medical screening and a health surveillance system must be established to determine the benzene exposure of those working at gasoline stations, with benzene concentrations in the ambient air at a 50 ppb setting point of action, especially for the urban areas.

## Figures and Tables

**Figure 1 ijerph-18-10014-f001:**
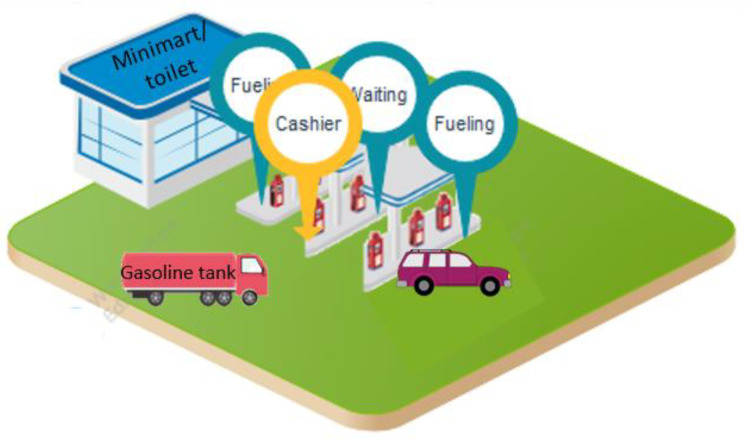
Gasoline station layout.

**Table 1 ijerph-18-10014-t001:** Adverse health effects of benzene exposure (*n* = 90).

Adverse Symptom	Had Symptom *n* (%)	Benzene >50% OEL Total = 3; *n* (%)	*p*-Value	tt-MA > BEI Total = 27; *n* (%)	*p*-Value
Low level (*n* = 54)					
Headache	40 (44.4)	2 (66.7) ^a^	0.583	14 (51.9) ^b^	0.355
Itchy skin/red rash/blisters	38 (42.2)	1 (33.3)	0.617	11 (40.7)	0.852
Exhaustion/fatigue	34 (37.8)	3 (100.0) ^a^	0.051	9 (33.3)	0.569
Dizziness	34 (37.8)	2 (66.7) ^a^	0.554	11 (40.7)	0.704
Nasal congestion	21 (23.3)	2 (66.7) ^a^	0.135	4 (14.8)	0.281
Sore throat/dry throat	17 (18.9)	0		5 (18.5)	0.953
Red eye/burning eyes/itchy eyes	15 (16.7)	1 (33.3)	0.425	2 (7.4)	0.215
Runny nose	12 (13.3)	2 (66.7) ^a^	0.046 *	4 (14.8)	0.747
Suffocation	11 (12.2)	1 (33.3)	0.327	7 (25.9)	0.015 *
Cough/hoarseness	11 (12.2)	1 (33.3)	0.327	6 (22.2)	0.058
Dry skin/cracked skin	8 (8.89)	1 (33.3)	0.246	2 (7.4)	1.000
Anorexia	7 (7.8)	1 (33.3)	0.218	4 (14.8)	0.191
Palpitations	3 (3.3)	0		1 (3.7)	1.000
Moderate level (*n* = 33)					
Chest pain	13 (14.4)	1 (33.3)	0.377	5 (18.5)	0.472
Numbness	10 (10.1)	1 (33.3)	0.098	3 (11.1)	1.000
Scurvy/Epistaxis/Bleeding	11 (12.2)	0		1 (3.7)	0.163
Insufficient/abnormal breathing	9 (10.0)	1 (33.3)	0.274	5 (18.5)	0.121
Nausea and vomiting	8 (8.9)	0		1 (3.7)	0.427
Blurred vision	6 (6.7)	0		0	-
Cramp	6 (6.7)	0		1 (3.7)	0.664
Confusion	3 (3.3)	0		1 (3.7)	1.000
Muscle weakness	3 (3.3)	0		2 (7.4)	0.213
Burning pain/swelling/wood skin	2 (2.2)	0		0	-
Tremor	2 (2.2)	0		2 (7.4)	0.088
Petechia	1 (1.1)	0		1 (3.7)	0.300
Depression	1 (1.1)	0		0	-
High level (*n* = 3)					
Anaemia	2 (2.2)	0		1 (3.7)	0.512
Unconsciousness	1 (1.1)	0		0	-

^a^ indicates that more than 50% of the total workers had been exposed to benzene >50% OEL; ^b^ indicates that more than 50% of the total workers had detected tt-MA > BEI; * indicates a significant association of the symptom with >50%OEL exposure by Fisher’s exact test (*p* ≤ 0.05).

**Table 2 ijerph-18-10014-t002:** Bivariate analysis of the relationship between personal factors and moderate to high levels of adverse effects (*n* = 151).

Characteristic	Total	Moderate to High Levels of Effects *n* (%)	Non-Symptomatic or Mild Symptoms *n* (%)	OR	95%CI	*p*-Value
Personal characteristics and health information						
Gender						
Male	56	13 (23.21)	43 (76.79)	0.95	0.43–2.06	0.889
Female	95	23 (24.21)	72 (75.79)	1.00		
Age (years)						
≤30	61	21 (34.43)	40 (65.57)	2.63	1.22–5.65	0.012 *
>30	90	15 (16.67)	75 (83.33)	1.00		
Education level						
Primary school and lower	89	24 (26.97)	65 (73.03)	1.54	0.70–3.37	0.276
Secondary school and higher	62	12 (19.35)	50 (80.65)	1.00		
BMI (kg/m^2^)						
Overweight and Obese	88	19 (21.59)	69 (78.41)	0.48	0.09–2.46	0.378
Underweight	12	2 (16.67)	10 (83.33)	0.66	0.30–1.45	0.303
Normal	51	15 (29.41)	36 (70.59)	1.00		
Income (baht/month)						
≥10,000	50	11 (22.00)	39 (78.00)	0.86	0.38–1.92	0.707
<10,000	101	25 (24.75)	76 (75.25)	1.00		
Congenital disease						
Yes	49	11 (22.45)	38 (77.55)	0.89	0.40–2.00	0.780
No	102	25 (24.51)	77 (76.49)	1.00		
Anaemia						
Yes	34	10 (29.41)	24 (70.59)	1.46	0.61–3.44	0.394
No	117	26 (22.22)	91 (77.78)	1.00		
Anaemia history						
Yes	24	5 (20.83)	19 (79.17)	0.81	0.28–2.36	0.703
No	127	31 (24.41)	96 (75.59)	1.00		
Annual health check-up						
No	121	30 (24.79)	91 (75.21)	1.32	0.49–3.53	0.576
Yes	30	6 (20.00)	24 (80.00)	1.00		
Workplace information						
Zone						
Urban area	28	10 (35.71)	18 (64.29)	1.00	0.37–2.71	1.000
Suburban area	81	11 (13.58)	70 (86.42)	0.28	0.12–0.69	0.006
Rural area	42	15 (35.71)	27 (64.29)	1.00		
Job function						
Fuelling	107	28 (23.93)	79 (76.07)	1.02	0.42–2.51	0.961
Cashier	34	8 (23.53)	26 (76.47)	1.00		
Working experience						
≥1 year	82	17 (20.73)	65 (79.27)	0.69	0.32–1.46	0.329
<1 year	69	19 (27.54)	50 (72.46)	1.00		
Shift work						
Day	98	20 (20.41)	68 (79.59)	0.59	0.28–1.27	0.183
Night	53	18 (30.19)	37 (69.81)	1.00		
Working days per week						
≥6 day	28	8 (28.57)	20 (71.43)	1.36	0.54–3.41	0.522
<6 day	123	28 (22.76)	95 (77.24)	1.00		
Working hours per day						
≥8 h	92	19 (20.65)	73 (79.35)	0.64	0.30–1.37	0.254
<8 h	59	17 (28.81)	42 (71.19)	1.00		
Use of uniform						
No	49	11 (22.45)	38 (77.55)	0.89	0.40–2.00	0.780
Yes	102	25 (24.52)	77 (75.49)	1.00		
Training experience						
No	41	19 (46.34)	22 (53.64)	4.72	2.12–10.54	<0.001 *
Yes	110	17 (15.45)	93 (84.55)	1.00		
Personal protective equipment (PPE) use						
Yes	72	23 (31.94)	49 (60.06)	2.38	1.10–5.17	0.025 *
No	29	13 (16.46)	16 (83.54)	1.00		
Whether participants were exposed to benzene in previous employment						
No	41	10 (24.39)	31 (75.61)	1.04	0.45–2.41	0.923
Yes	110	26 (23.64)	84 (76.36)	1.00		
Shower after work						
No	53	14 (26.42)	39 (73.58)	1.24	0.57–2.69	0.587
Yes	98	22 (22.45)	72 (71.55)	1.00		
Hand washing						
No	21	8 (38.10)	13 (61.90)	2.24	0.85–5.94	0.114
Yes	130	28 (21.54)	102 (78.46)	1.00		
Eat food in the working area						
Yes	119	33 (27.73)	86 (72.27)	3.71	1.06–13.01	0.019 *
No	32	3 (9.38)	29 (90.62)	1.00		
Eat food during work						
Yes	120	35 (29.17)	85 (70.83)	12.35	1.62–94.14	<0.001 *
No	31	1 (3.23)	30 (96.77)	1.00		

* Significant factors of correlation with adverse effect of benzene exposure at *p*-value < 0.05.

**Table 3 ijerph-18-10014-t003:** Multiple logistic regression analysis of the potential risk factors of moderate and high levels of adverse health effects among gasoline station workers (*n* = 151).

Characteristic	Adverse Effect *n* (%)	OR	OR_adj_	95%CI	*p*-Value
Gender					
Male	13 (23.21)	0.95	1.33	0.56–3.19	0.517
Female	23 (24.21)	1.00	1.00		
Age (year)					
≤30	21 (34.43)	2.63	1.96	0.84–4.55	0.119
>30	15 (16.67)	1.00	1.00		
Education level					
Primary school or lower	24 (26.97)	1.54	1.76	0.73–4.25	0.209
high school or higher	12 (19.35)	1.00	1.00		
Zone					
Urban area	10 (35.71)	1.00	1.93	0.57–6.51	0.479
Suburban area	11 (13.58)	0.28	0.671	0.22–2.02	0.287
Rural area	15 (35.71)	1.00	1.00		
Shift work					
Day	16 (30.19)	0.59	0.56	0.22–1.39	0.214
Night	20 (20.41)	1.00	1.00		
Training					
No	19 (46.34)	4.72	5.22	2.16–12.58	<0.001 *
Yes	17 (15.45)	1.00	1.00		
PPE use					
Yes	23 (31.94)	2.38	2.11	0.86–5.16	0.101
No	13 (16.46)	1.00	1.00		
Hand washing					
No	8 (38.10)	2.24	1.84	0.57–5.98	0.312
Yes	28 (21.54)	1.00	1.00		
Eat food in the working area					
Yes	33 (27.73)	3.71	0.64	0.009–4.28	0.648
No	3 (9.38)	1.00	1.00		
Eat food during work					
Yes	35 (29.17)	12.35	16.08	1.96–131.74	0.010 *
No	1 (3.23)	1.00	1.00		

Remark: Gender, age, education level was always included as the confounders. * Significant factors effecting with health adverse effect of benzene exposure *p*-value < 0.05.

## Data Availability

Data availability upon personal request.

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
