# Peer review of "Factors Affecting Adverse Health Effects of Gasoline Station Workers"

_ijerph, 2021, doi:10.3390/ijerph181910014_

Round 1

Reviewer 1 Report

This manuscript is an interesting and good study. However, it will be a better paper if you describe the manuscript in more detail and add a review.

In the Abstract part, the author said that the benzene concentration exceeds NIOSH REL. However, the concentrations evaluated with area sampling cannot be directly compared to OELs. 
Therefore, write a sentence considering the area sample or estimate the concentration considering the actual worker's movement during working hours and compare it.

IARC is a very important document on health effects from exposure to hazardous substances (especially carcinogens such as benzene). However, there is little mention of IARC documents in this manuscript. Therefore, please briefly mention the health effects in the IARC documents in this paper.

Did only benzene affect adverse health effects? What is the health impact of gasoline (VOCs) exposure? Please discuss it in Discussion section.

To improve the completeness of the paper, add a picture of a representative gas station to check the number of gas stations, the location of gasoline tanks, the location of offices, and the waiting location of workers.

The high concentration at the measurement location and the high exposure of the worker are different. (The concentration of benzene varies greatly depending on the location of the measurement because the gas vapour diffuses quickly.) In this study, the benzene concentration was evaluated using the area sampling method. How many samples did you measure at each gas station? Where did you measure the sample in the gas station? Please add a new table about on benzene concentrations in the air.

This reviewer is conducting a work-related assessment of leukemia from a gas station. It's great if you can provide the estimated cumulative exposure of 151 workers to benzene. ex mean 5 ppm/years (0.01~10 ppm/years) (In particular, the cumulative exposure levels of workers with severe symptoms.) (optional)

Thank you.

Author Response

Reviewer 1

comments

  1. In the Abstract part, the author said that the benzene concentration exceeds NIOSH REL. However, the concentrations evaluated with area sampling cannot be directly compared to OELs.
    Therefore, write a sentence considering the area sample or estimate the concentration considering the actual worker's movement during working hours and compare it.

Response: Corrected in line 19 “Area samplings for benzene concentration”

Explanation: Area samplings for benzene concentration by setting the collecting sample in 2-3 setting near gasoline dispenser (service point), waiting service area (near dispenser as well) and cashier booth or cashier working area in all 41 stations, where workers almost time moving around the place during working hours, and it was calculated to 8 hour or working hours to compare to OELs.

  1. IARC is a very important document on health effects from exposure to hazardous substances (especially carcinogens such as benzene). However, there is little mention of IARC documents in this manuscript. Therefore, please briefly mention the health effects in the IARC documents in this paper.

Response: Added in line 75 “carcinogenic risks to human from low exposure but in long term which recommend by IARC (acute myeloid leukemia, acute lymphocytic leukemia, multiple myeloma, and non-Hodgkin lymphoma) [17]”

Reference No.  17 “IARC. Benzene, Monographs on the evaluation of carcinogenic risks to humans; Lyon: FRANCE, 2018.

  1. Did only benzene affect adverse health effects? What is the health impact of gasoline (VOCs) exposure? Please discuss it in Discussion section.

Response: Added in line 336 “Most of worker had the symptoms in mild level (such as headache, itchy skin, fatigue, dizziness, etc.) it was consistent with previous study that found working at gasoline stations was exposed to the VOCs and BTEX substance. Most acute symptoms associated with irritated of the respiratory tract and skin [21, 22]. But this study found some workers who had tt-MA level in urine after work more than BEI standard also had the symptom report for hematopoietic effect (anemia history). Therefore, we still found the adverse health effect from benzene exposure in gasoline station workers”

Ref No. 23-24.

  1. Chaiklieng, S; Suggaravetsiri, P; Kaminskic, N; Autrup, H. Exposure to benzene and toluene of gasoline station workers in Khon Kaen, Thailand and adverse effects. HERA 2020, 27(7), 1823-1837.
  2. Kuranchie, FA; Angnunavuri, PN; Attiogbe, F; Nerquaye-Tetteh, EN. Occupational exposure of benzene, toluene, ethylbenzene and xylene (BTEX) to pump attendants in Ghana: Implications for policy guidance. Cogent Environ. Sci. 2019, 5.doi:org/10.1080/23311843.2019.1603418 1603418

  1. To improve the completeness of the paper, add a picture of a representative gas station to check the number of gas stations, the location of gasoline tanks, the location of offices, and the waiting location of workers.

Response:

  • This study there was no include the office location because the fueling and cashier workers work at service station near fueling dispenser.
  • The example of gasoline station layout:

Figure 1. Gasoline station layout

  1. The high concentration at the measurement location and the high exposure of the worker are different. (The concentration of benzene varies greatly depending on the location of the measurement because the gas vapour diffuses quickly.) In this study, the benzene concentration was evaluated using the area sampling method. How many samples did you measure at each gas station?

Where did you measure the sample in the gas station? Please add a new table about on benzene concentrations in the air.

Response: We set the sampling point for 2-3 points per station as follows: 1. The point near gasoline dispenser (service point) 2. Waiting service area (near dispenser as well) 3. Cashier booth or cashier working area.  Added in line 148

Parameters

Benzene concentration (ppb)

Median (min-max)

4.6 (0.1 - 136.9)

95th percentile

32.2

IQR (25th – 75th percentile)

10.05 (2.0 - 12.05)

% OEL exceed (100 ppb) N (%)

1 (0.66)

  1. This reviewer is conducting a work-related assessment of leukemia from a gas station. It's great if you can provide the estimated cumulative exposure of 151 workers to benzene. ex mean 5 ppm/years (0.01~10 ppm/years) (In particular, the cumulative exposure levels of workers with severe symptoms.) (optional)

Response:  

Added in line 320 “The result of this study shows that the workers who had more working experience had accumulative benzene concentration higher than some workers who was working less than 3 years. (The data not normality distribution so we should consider for IQR and 95th percentile). The accumulated concentration for workers who had the experience of work in gasoline station for 5 year and more is 67.6 ppm (95th percentile). The estimate of benzene concentration exposure accumulation per year had the IQR (25th Percentile – 75th Percentile) is 2.89 (0.57 – 3.47) ppm and 95th Percentile is 9.27 ppm. From this estimate calculation of benzene exposure per year it was shown that the worker had risk for hematopoietic cancer consistent with previous study shown the accumulation benzene concentration from 0.5 – 1 ppm per year [21]

Ref: Yoon, J; Kwak, W; Ahn, Y. A brief review of relationship between occupational benzene exposure and hematopoietic cancer. AOEMJ 2018, 30,33.

Reviewer 2 Report

Review of “Factors affecting adverse health effects of gasoline station workers”
The objective of this manuscript is to identify factors associated with adverse health effects from benzene exposure among gasoline workers in Thailand. A survey was administered to gather exposure-related information. Air sampling and biological exposure monitoring for benzene were conducted to characterize the benzene exposure levels among the gasoline workers. The methods are sound, and the writing is clear. I would like to offer some comments for the authors to improve the manuscript.
1.    Abstract
•    Page 1, line 19: Indicate whether personal air sampling or area sampling method was used.
•    Page 1, line 24: Please note that the NIOSH REL is 0.1 ppm for benzene. The REL is based on exposures assumed for 8 hours a day, 250 days/year for a 30-year working lifetime. Therefore, in order to compare the monitored benzene exposure level to REL, the authors have to calculate the 8-hour time-weighted exposure level.
When the authors stated that “the detected air benzene reached an amount higher than the NIOSH REL”, it is not appropriate to make this comparison. The authors need to calculate the 8-hour TWA, then compare those numbers to REL.
For example, if the sampling duration is 4 hours, the measured exposure level is 8 ppm, then the 8-hour TWA concentration is 8 ppm * 4 hours/ 8 hours = 4 ppm
2.    Background
•    Page 2, line 58: Is the OEL of 0.1 ppm legally enforceable in workplaces in Thailand. If so, please indicate that this is a regulated OEL in Thailand.
3.    Materials and Methods
•    Page 2 line 99 and page 3 line 104: Please indicate if the gasoline workers work by shift and how many hours they work per day.
•    I suggest in the method section, the authors should add a section of adjusted occupational exposure limit. Due to the extended work hours of gasoline workers, the authors need to use the Brief and Scala model to calculate adjusted OEL. The calculation method is recommended by the The American Conference of Governmental Industrial Hygienists and can be found here: https://www.ccohs.ca/oshanswers/hsprograms/occ_hygiene/occ_exposure_limits.html
•    Air sampling: It is not clear if the authors conducted area or personal air monitoring. If personal air monitoring was conducted, please explain in the air sampling method section. In addition, the authors need to provide the make and model of the air sampling equipment and which lab conducted the sample analysis.
•    Detection of tt-MA in Urine after work: Please provide that the samples were collected at the end of the work shift.
4.    Results
•    It is not clear to me whether the authors calculated the 8-hour TWA exposure levels. The authors need to calculate the 8-hour TWA before comparing it with OEL.
•    Since the adjusted OEL must be used for comparison, the authors may need to re-do the analysis for Table 1.

Author Response

Reviewer 2

Comments

  1.  Abstract
    •    Page 1, line 19: Indicate whether personal air sampling or area sampling method was used.

Response: Added in line 19 “Area samplings for benzene concentration”

  •    Page 1, line 24: Please note that the NIOSH REL is 0.1 ppm for benzene. The REL is based on exposures assumed for 8 hours a day, 250 days/year for a 30-year working lifetime. Therefore, in order to compare the monitored benzene exposure level to REL, the authors have to calculate the 8-hour time-weighted exposure level.
    When the authors stated that “the detected air benzene reached an amount higher than the NIOSH REL”, it is not appropriate to make this comparison. The authors need to calculate the 8-hour TWA, then compare those numbers to REL.
    For example, if the sampling duration is 4 hours, the measured exposure level is 8 ppm, then the 8-hour TWA concentration is 8 ppm * 4 hours/ 8 hours = 4 ppm

Response: This study uses the concentration from calculated for 8 hours per day, although we have air sampling for collected the benzene concentration only 4 hours.

  1.    Background
    •    Page 2, line 58: Is the OEL of 0.1 ppm legally enforceable in workplaces in Thailand. If so, please indicate that this is a regulated OEL in Thailand.

Response “Legally enforceable in workplaces in Thailand is 1.00 ppm (OEL)”

Ref. Announcement of the Department of Labor Protection and Welfare Re: Limit of Hazardous Chemical Concentration, 2017.

  1.    Materials and Methods
    •    Page 2 line 99 and page 3 line 104: Please indicate if the gasoline workers work by shift and how many hours they work per day.

Response – Urban area: service time 5 am – 10 pm, 2 shifts: 8-10 hrs/ shift

  • Suburban: 24 hrs: 3 shifts, 7-8 hrs/ shift
  • Rural: 5 am - 8 pm: 7-8 hrs/ shift

Added in line 102

  •    I suggest in the method section, the authors should add a section of adjusted occupational exposure limit. Due to the extended work hours of gasoline workers, the authors need to use the Brief and Scala model to calculate adjusted OEL. The calculation method is recommended by the The American Conference of Governmental Industrial Hygienists and can be found here: https://www.ccohs.ca/oshanswers/hsprograms/occ_hygiene/occ_exposure_limits.html

Response: This study uses the concentration from calculated for 8 hours per day. Although, we had air sampling for collected the benzene concentration only 4 hours. Therefore, we had confirmed for the concentration of benzene that we use in this study.

  •    Air sampling: It is not clear if the authors conducted area or personal air monitoring. If personal air monitoring was conducted, please explain in the air sampling method section. In addition, the authors need to provide the make and model of the air sampling equipment and which lab conducted the sample analysis.

Response:

This study uses the area sampling for collected benzene concentration because of for safety practice reason, personal pump with battery cannot be close to the dispenser during monitoring. However, area samples could be representative average from average concentration. Using NIOSH No.1501 standard methods. For analysis we use the standard lab had certified of ISO/ IEC 17025: 2005 (Ref No. 0303/17133) Added in line 148.

  •    Detection of tt-MA in Urine after work: Please provide that the samples were collected at the end of the work shift.

Response: Added in line 162 “A 30 ml amount of spot urine was collected at the end of shift work”

  1.  Results
    •    It is not clear to me whether the authors calculated the 8-hour TWA exposure levels. The authors need to calculate the 8-hour TWA before comparing it with OEL.

Response: This study uses the concentration from calculated for 8 hours per day, although we have air sampling for collected the benzene concentration only 4 hours. Therefore, we had confirmed for the concentration of benzene that we use in this study.

  •    Since the adjusted OEL must be used for comparison, the authors may need to re-do the analysis for Table 1.

Response: The reason is following above.

Reviewer 3 Report

Summary

This study investigated the effects of exposure to benzene (and indirectly other VOCs) on the health of workers are gasoline stations. 151 gasoline station workers from 41 gasoline stations in Khon Kaen province in Thailand completed a questionnaire detailing the range of health problems experienced that could possibly be linked with exposure to benzene. Spot urine tests for tt-muconic acid (tt-MA) were conducted on the workers. Air samples were collected at each gasoline station to measure ambient benzene concentrations. It was found that fuelling workers experience adverse symptoms more frequently than do cashiers. The incidence of symptom occurrence was documented. Having no safety training and eating while working were identified to be significant risk factors associated with more severe symptoms.

I feel this study makes a valuable contribution to an extremely severe environmental health problem, i.e. the health of gasoline station workers as a result of exposure to high levels of VOCs. The sampling and data analysis are competent and the paper is well written.

Major Issues

There are no major issues to be addressed in this paper.

Minor Issues

Please address the following:

  1. Abstract: I recommend that the country and province where the investigation was conducted be stated in the abstract to give the reader context.

  1. Lines 38-42: I am not sure what to make of the two introductory sentences. Car ownership is increasing but fuel use declining in Thailand? Please state the point clearly here.

  1. Line 48: avoid colloquial expressions like ‘pretty harmful’.

  1. Line 136, section 2.4: Please can a little more information on the sampling be provided. How many samples were collected per site? At what time of day was the sampling done? Where were the samples collected, i.e. how representative are they of exposure level?

  1. Lines 192-193: If 61 out of 151 participants were younger than 30, then it is not true that ‘most’ of them are younger than 30. Please reword.

  1. Line 198: Please can the US dollar-equivalent of the monthly income be stated in brackets for international readers.

  1. Line 308: Please clarify what ‘low symptoms’ are.

  1. Line 332: Correct spelling of ‘benzine’.

  1. Line 337: Correct: ‘2.63 times more likely’

  1. References: The first four references need to include the author or institution that is the source of the data.

Author Response

Reviewer 3

Comments

Please address the following:

  1. Abstract: I recommend that the country and province where the investigation was conducted be stated in the abstract to give the reader context.

Response: Added in line 17 “in Khon Kean province, Thailand”

  1. Lines 38-42: I am not sure what to make of the two introductory sentences. Car ownership is increasing but fuel use declining in Thailand? Please state the point clearly here.

Response: Added in line 42 “the sales figures of gasoline stations from 2012–2015 showed that gasoline sales had increased from the previous years”

  1. Line 48: avoid colloquial expressions like ‘pretty harmful’.

Response: Corrected in line 48 “Benzene released from the fuel vapours can be harmful when inhaled”

  1. Line 136, section 2.4: Please can a little more information on the sampling be provided. How many samples were collected per site? At what time of day was the sampling done? Where were the samples collected, i.e. how representative are they of exposure level?

Response: We were collected 2-3 sampling of each stations. Because some stations, cashier positions do not work in closed booths, therefore store the payer to place a cashier or a table in the middle of the island, representing the cashier, cashier (34 workers), and the other part is a position for fuelling and waiting location.

  1. Lines 192-193: If 61 out of 151 participants were younger than 30, then it is not true that ‘most’ of them are younger than 30. Please reword.

Response: Rewording in line 206 “Most of them were younger than 30 years (n= 90; 59.6%)”

  1. Line 198: Please can the US dollar-equivalent of the monthly income be stated in brackets for international readers.

Response: Added in line 212 “10,634.5 ± 3407.0 Thai Baht (ranging between 9000 and10,000 Thai Baht/ 300 US dollar)”

  1. Line 308: Please clarify what ‘low symptoms’ are.

Response:  Low level symptoms such as headache, Exhaustion/ fatigue, Dizziness, Nasal congestion, Sore throat / dry throat, Red eye / burning eyes /itchy eyes, Runny nose, Cough / hoarseness, Dry skin/ cracked skin.

  1. Line 332: Correct spelling of ‘benzine’.

Response: Corrected “This finding suggests the possibility of workers experiencing a chronic or long-term exposure to benzene.”

  1. Line 337: Correct: ‘2.63 times more likely’

Response: Corrected “Those who were younger than 30 years were 2.63 times more likely to report symptoms related to benzene exposure than their older peers”

  1. References: The first four references need to include the author or institution that is the source of the data.

Response: Corrected

  1. Department of land transport. Number of new registered cars. Available online: https://web.dlt.go.th/statistics (accessed on 20 October 2020)
  2. Department of Energy Business. The amount of fuel sold per day. Available online: https://www.doeb.go.th/info/value_fuel.php. (accessed on 5 December 2019)
  3. Department of Energy Business. Determine the characteristics and quality of benzene 2019. Available online: http://elaw.doeb.go.th. (accessed on 6 December 2019)
  4. Department of Energy Business. Determine the characteristics and quality of benzene 2012. Available online: http://elaw.doeb.go.th (accessed 6 December 2019)

Round 2

Reviewer 1 Report

The manuscript has been revised well.
It's enough to be published in this journal.

Thank you.

Reviewer 2 Report

Thank you the authors for addressing my comments.

This manuscript is a resubmission of an earlier submission. The following is a list of the peer review reports and author responses from that submission.

Round 1

Reviewer 1 Report

The main concern regarding the paper is that the symptoms appear to be reported by the subjects in terms of answers reported in a questionnaire. Many of the symptoms, notably moderate and high-level effects (e.g., petechia, anemia, depression) would require a medical diagnosis. The use of medical records in this study should be clarified. It is also assumed that all of the symptoms are due to benzene exposure. It should be noted that gasoline is primarily a mixture of aliphatic hydrocarbons that may have contributed to the symptoms observed.

Minor points:

            Line 20: Change to “biomarker of benzene exposure”

            Line 142 and throughout: for tt-MA analysis, use mg/g CRN; i.e., abbreviate creatinine   as CRN to avoid confusion with the element chromium (Cr).

            Table 2: Change to “Workplace information”

Author Response

Response to Reviewer 1 Comments

Point 1: Many of the symptoms, notably moderate and high-level effects (e.g., petechia, anemia, depression) would require a medical diagnosis.

Response 1: This is the limitation of this study; we got the report of their health symptom experience during working with benzene exposure by interview. But the symptom of high level (level 4) i.e., anemia and unconsciousness were confirmed by their medical treatment history.

Added information to line 131 and limitation of study was added in line 418.

Point 2: The use of medical records in this study should be clarified. It is also assumed that all of the symptoms are due to benzene exposure.

Response 2: For the symptoms from benzene exposure and toxicity following the experience symptoms found form the previous study (Chaiklieng et al., 2015; 2021) for the reference and in the moderate to severe level is the specific symptom from benzene exposure.

Point 3: It should be noted that gasoline is primarily a mixture of aliphatic hydrocarbons that may have contributed to the symptoms observed.

Response 3: For the other mixture substance, we also classified for the co-health adverse symptom in mild level such as headache, sore throat, eye skin irritant, dizziness (Tunsaringkarn et al., 2012). Added to line 51-57.

Minor point

Point 1: Line 20: Change to “biomarker of benzene exposure

Response 1: Changed from “biomarker of benzene detection to biomarker of benzene exposure” Abstract in Line 20.

Point 2: Line 142 and throughout: for tt-MA analysis, use mg/g CRN, i.e., abbreviate creatinine   as CRN to avoid confusion with the element chromium (Cr).

Response 2: Changed all Cr to CRN

Point 3: Table 2: Change to “Workplace information”

Response 3: Changed working information to workplace information in Table 2

Point 4: Please provide us with a blank copy of informed consent

Response 4: The informed consent was attached.

Point 5: Please explain * in table footer, for tables 3 and 4

Response 5: Footer Table 2: * Significant factors correlation with health adverse effect of benzene exposure at p-value < 0.05

          Footer Table3: * Significant correlation with health adverse effect of benzene exposure at p-value < 0.05

Point 6: Please check references cations and duplicates, as it has been observed that ref 3 and 4 are duplicate (all references must be cited in order through the main text)

Response 6: Reference 3 and 4 had different year of information but they from the same source: Reference 3 is the information citation of 2019; Reference 4 is the information citation of 2012. Added 2019 year in reference 3 and 2012 in reference 4.

Reviewer 2 Report

This is a cross-sectional study of self-reported symptoms and urinary biomarkers of benzene exposure in 151 gasoline station workers at 41 gasoline stations in Khon Kaen province in Thailand, where automobile ownership and use is increasing dramatically. Air sampling levels and end of shift urinary tt-MA levels were elevated on average. Absence of safety training and eating during work hours were associated with symptom reporting. The authors recommend more training and restrictions on eating in the workplace. This is an interesting study with many important measurements. I offer some considerations for the authors in specific comments below.

Specific comments

  1. Gasoline is a complex mixture of hydrocarbons. While benzene is a very hazardous component, it is a small proportion of the mixture. The review in the Introduction focuses on benzene, but the most important health effects of benzene are longer latency than would be captured by a symptom questionnaire. There is insufficient coverage of the health effects of the other VOCs in gasoline.
  2. The sample size calculations on lines 81 and following are difficult to understand. Was it based on a P0 of 10% and a specified effect size of OR = 2.74? It would be good to specify the minimum ORs that could be detected as significant with a type 2 error rate of 0.20, rather than the very high power of 95% that was mentioned. There is inconsistency in these sentences for how references are cited (e.g., name, year vs. [#]).
  3. The sampling design (lines 88 and following) is complicated. This has implications for the analysis. Workers were purposively sampled from each station, so these observations are not independent, with persons clustered in places.
  4. Were the subjects interviewed or was the questionnaire self-administered (lines 103-4).
  5. In section 2.3 the authors use the term “symptom” very non-specifically and inaccurately. Leukemia is a disease, not a symptom. Anemia is measured with laboratory tests and is not a symptom. Rash and dry skin are signs, not symptoms. The categorization is also unjustified. For example, suffocation seems like more than a mild symptom. A quantitative measurement approach to the grouping of these symptoms could have been used (depending on the theory and motivation, EFA, PCA, or LCA could be used).
  6. Section 2.4: what were the sampling durations?
  7. Section 2.5: what quality control was used? Were measurements performed in duplicate? What were the relevant coefficients of variation? What proportion of measurements were < LOD and how were these values handled?
  8. Section 2.6:
    1. Is an “abnormal” distribution a non-normal distribution?
    2. The Fisher’s exact text is used for 2x2 tables. So tt-MA and air benzene levels were made binary for this analysis? Why and how? A better approach would be to handle the continuous variables as continuous while allowing for non-linearity.
    3. The analysis is not entirely clear. Did the author’s divide symptoms into two groups (moderate + severe/high vs. lower levels [what was the reference group?])? How was the clustering of workers in stations accounted for in the analysis? How was the clustering of stations in urban, suburban, and rural areas approached?
    4. How was BMI divided into these categories?
    5. How was multiple testing addressed?
  9. The tt-MA levels were very interesting. It would be good to know what the adjusted predictors of levels were. The tt-MA levels were much higher than would be expected given the air exposure levels. Only 2 workers were exposed above the NIOSH REL but 27% exceeded the ACGIH BEI, in some cases by a factor of 10 or more. Even the cashiers had levels that were 3X the BEI. How is this explained? Was urinary cotinine measured?
  10. Section 3.3: what does the attribution of symptoms by workers to benzene exposure mean (“…reported having symptoms associated with benzene exposure”)? How was anemia ascertained? It is not a symptom, it is measured by identifying low hemoglobin or hematocrit levels.
  11. Table 1: Many of these symptoms are associated with VOC exposure. How can attribution to benzene be made? Tables should be self-explanatory. I assume these are all unadjusted associations. Why is the denominator in this table 90? Shouldn’t it be 151?
  12. Section 3.4: There are many instances of methods in this section that should be moved to the appropriate subsection of methods.
  13. Table 2: Persons who used PPE were MORE likely to report symptoms? How is this explained? What does “experience of working with benzene” mean? These workers did not work directly with benzene, they worked near or around gasoline.
  14. Discussion: The limitations of the study should be more clearly addressed.
  15. Discussion: How was same source bias considered and addressed? Wouldn’t workers without training be more likely to report symptoms? The significant unadjusted associations in Table 2 suggest that the young, the untrained, those eating in the workplace, and those wearing PPE were more likely to report symptoms. This seems like same source bias and reporting bias. It also seems possible that workers at the same stations may have similarly reported symptoms because they discuss their concerns with others at the site, but site was not adjusted for and clustering was not accounted for.

Author Response

Response to Reviewer 2 Comments

Point 1: Gasoline is a complex mixture of hydrocarbons. While benzene is a very hazardous component, it is a small proportion of the mixture. The review in the Introduction focuses on benzene, but the most important health effects of benzene are longer latency than would be captured by a symptom questionnaire. There is insufficient coverage of the health effects of the other VOCs in gasoline.

Response 1: The suggestion was helpful to this study; the researcher add more symptom to the introduction part for cover all health effect from gasoline workers symptom report in line 54

“The most common symptom reported in gasoline station workers were headache, fatigue, throat irritation, nose irritation, nausea, dizziness, and depression [10]” From the literature review, we found the main hazard chemical from gasoline is benzene (IARC 1989), so we focus and show only harmful on benzene to gasoline station workers.

Point 2: The sample size calculations on lines 81 and following are difficult to understand. Was it based on a P0 of 10% and a specified effect size of OR = 2.74? It would be good to specify the minimum ORs that could be detected as significant with a type 2 error rate of 0.20, rather than the very high power of 95% that was mentioned. There is inconsistency in these sentences for how references are cited (e.g., name, year vs. [#]).

Response 2: Line 90: The reference rate was a tt-MA level of more than 10% BEI in persons who had no job training (Odd ratio = 2.74) Chaiklieng et al., 2019.

         Added information in line 91 “from the study of factors effecting urinary tt-MA detection among benzene exposed workers at gasoline station”

Point 3: The sampling design (lines 88 and following) is complicated. This has implications for the analysis. Workers were purposively sampled from each station, so these observations are not independent, with persons clustered in places.

Response 3: The sample size was allocated to urban, suburban, and rural area of gasoline station in Khon Kaen as shown in the figure below. From the sampling station according to the distributed zone, 3-4 workers were participated as volunteer and representative of the station.

Point 4: Were the subjects interviewed or was the questionnaire self-administered (lines 103-4).

Response 4: The subjects were interviewed by the researcher. Corrected in line 113: Participants were interviewed by researchers about the symptoms they had experienced in the past three months.

Point 5: Section 2.3 In section 2.3 the authors use the term “symptom” very non-specifically and inaccurately. Leukemia is a disease, not a symptom.

Response 5: We are accepted and would like to thank you for this suggestion of leukemia that not symptom. In this study did not get leukemia report from subject.

Point 6: Section 2.3 Anemia is measured with laboratory tests and is not a symptom.

Response 6: For the anemia we found the information by interview for 2 workers and they confirmed their information after they got diagnosis by physician before. Added the important criteria for justified to health adverse effect of level 4 and 5 by physician in line 131

Point 7: Section 2.3 Rash and dry skin are signs, not symptoms. The categorization is also unjustified. For example, suffocation seems like more than a mild symptom. A quantitative measurement approach to the grouping of these symptoms could have been used (depending on the theory and motivation, EFA, PCA, or LCA could be used).

Response 7: There was no report from the previous study for the suffocation is the specific symptom form benzene exposure. This symptom can be caused find from mixture substance in gasoline station workers that have report in the previous study (Chaiklieng et al., 2015, 2021). Therefore, we were classified this symptom to mild level.

Point 8: Section 2.4: what were the sampling durations?

Response 8: June - August 2018; At daytime and night-time.

Point 9: Section 2.5: what quality control was used? Were measurements performed in duplicate? What were the relevant coefficients of variation? What proportion of measurements were < LOD and how were these values handled?

Response 9: The tt-MA was analysed by HPLC which used LOD <0.001 mg/g Creatinine and all urine sample (100%) result had tt-MA higher than LOD; min-max: 0.005 – 5.99 mg/g Creatinine.

Point 10: Section 2.6 Is an “abnormal” distribution a non-normal distribution?

Response 10: Yes, Abnormal = non normal distribution. Changed from abnormal to non-normal in line 162.

Point 11: Section 2.6 The Fisher’s exact text is used for 2x2 tables. So tt-MA and air benzene levels were made binary for this analysis? Why and how? A better approach would be to handle the continuous variables as continuous while allowing for non-linearity.

Response 11: tt-MA and air benzene does not include to logistic regression model in this study, but they were compared to the standard recommendation as OEL, BEI for analysis of symptom as shown in Table 1. Added to statistical analysis in line 154 “STATA version 10.0 software was used to analysis the symptom indicated the workers had been exposed to benzene >50% OEL (>50 ppb) and workers had detected tt-MA > BEI (>500 µg/g CRN) by Chi-square and Fisher’s exact test significantly considering  at p-value ≤0.05.”

Point 12: Section 2.6 The analysis is not entirely clear. Did the author’s divide symptoms into two groups (moderate + severe/high vs. lower levels [what was the reference group?])? How was the clustering of workers in stations accounted for in the analysis? How was the clustering of stations in urban, suburban, and rural areas approached?

Response 12: Yes, the outcome of this study is 0 (reference group) = no symptom and mild symptom level, 1 = moderate and high symptom level

We are classifying the workers into 2 group is fueling workers who is the main responsible to put in gasoline, and cashier worker who is the main responsible to the accountant.

We are clustering of gasoline station into 2 group is 1) urban area (gasoline stations beside Mittraphap Road and those no more than 2 kilometres away from it), 2) suburban area (gasoline stations that are between 2 kilometres and 5 kilometres away from Mittraphap Road), and and  the rural area (gasoline stations that are at least 5 kilometres away from Mittraphap Road) (Chaiklieng at all., 2019).

Point 13: Section 2.6 How was BMI divided into these categories?

Response 13: We were divided BMI into 2 group is 1) Normal = 18.5 – 22.9 kg/m2 2) Abnormal (underweight, normal, overweight, and obese) = less than 18.5 and more than 22.9 kg/m2

Point 14: Section 2.6 How was multiple testing addressed?

Response 14: Multiple logistic regression analysis included factors with p<0.25 and the confounders of gender, age, and education level, the significant factors presented with ORadj, 95%CI at p<0.05 were from the initial model. Added information to line 175.

Point 15: The tt-MA levels were very interesting. It would be good to know what the adjusted predictors of levels were. The tt-MA levels were much higher than would be expected given the air exposure levels. Only 2 workers were exposed above the NIOSH REL but 27% exceeded the ACGIH BEI, in some cases by a factor of 10 or more. Even the cashiers had levels that were 3X the BEI. How is this explained? Was urinary cotinine measured?

Response 15: For cashier workers was similar working condition with fueling workers which confirmed form the previous study (Chaiklieng et al., 2021 “based on similar characteristics, such as refueling services handled by refueling workers, and cashiers working in a one-side-opened booth located in the center of the station, which is not that far from the dispenser nozzles. Moreover, some of the refueling workers do cashier tasks or supporting work at the cashier’s desk. That might explain the similar concentrations of benzene and toluene found in the working environment of refueling workers and cashiers”). This study does not measure cotinine in urine.

Point 16: Section 3.3: what does the attribution of symptoms by workers to benzene exposure mean (“…reported having symptoms associated with benzene exposure”)? How was anemia ascertained? It is not a symptom; it is measured by identifying low hemoglobin or hematocrit levels. 080652

Response 16: The limited of this study, we do not measure the high and higher symptom by medical diagnosis. But we confirm the information that got from subjects by their medical history treatment. Added the criteria for the information of high and severe health adverse effect symptom in the method section 2.3

Point 17: Table 1: Many of these symptoms are associated with VOC exposure. How can attribution to benzene be made? Tables should be self-explanatory. I assume these are all unadjusted associations. Why is the denominator in this table 90? Shouldn’t it be 151?

Response 17: For the Table 1. We need to show the symptom from gasoline station workers were report only 90 persons.

Point 18: Section 3.4: There are many instances of methods in this section that should be moved to the appropriate subsection of methods.

Response 18: Cut the detail of how to bring the factor from univariate to the multivariate analysis in line 265 “with p < 0.05 among those factor”

Point 19: Table 2: Persons who used PPE were MORE likely to report symptoms? How is this explained? What does “experience of working with benzene” mean? These workers did not work directly with benzene, they worked near or around gasoline.

Response 19: We are explained in discussion “The workers reported using PPE, i.e., wearing hand coverings and masks made of cotton or fabric to protect against sunlight, which might increase the accumulation of benzene into the body.” experience of working with benzene is worker have ever been working past job which exposed to benzene before such as garage, shoes factory, etc.)

Point 20: Discussion: The limitations of the study should be more clearly addressed.

Response 20: This study had the limitation for the confirmation of health adverse effect for high and severe level (level 4-5) should have diagnosis or laboratory testing. Added the limitation of this study to discussion in line 418.

Point 21: Discussion: How was same source bias considered and addressed? Wouldn’t workers without training be more likely to report symptoms? The significant unadjusted associations in

Table 2 suggest that the young, the untrained, those eating in the workplace, and those wearing PPE were more likely to report symptoms. This seems like same source bias and reporting bias. It also seems possible that workers at the same stations may have similarly reported symptoms because they discuss their concerns with others at the site, but site was not adjusted for and clustering was not accounted for.

Response 21: We were distributed of collecting data to urban, sub urban, and rural area of Khon Kaen province so we can defense about addressed bias.

In this study found the significant factor were strong associated with benzene exposure: working without training and eating food during work.

  • Working without training is consistent with the previous study in gasoline station worker’s same condition (Chaiklieng et al., 2019). This factor it’s can be led to the other risk factor coming to gasoline station workers
  • Eating food during work is the risk behavior and personal hygiene because the nature of north east region of Thailand people prefer to eat with their hands. The main food of this region they like to eat the sticky rice by using their hand. This reason made these workers had more risk of ingesting benzene in addition to breathing.

Round 2

Reviewer 2 Report

I thank the authors for their reply. My concerns about sources of bias, problems with statistical analysis, and sampling design remain.